

# Using machine learning analysis to interpret the relationship between music emotion and lyric features

Liang Xu[1], Zaoyi Sun[2], Xin Wen[1], Zhengxi Huang[1], Chi-ju Chao[3] and Liuchang Xu[4,5]

[1] Department of Psychology and Behavioral Sciences, Zhejiang University, Hangzhou, China
[2] College of Education, Zhejiang University of Technology, Hangzhou, China
[3] Department of Information Art and Design, Tsinghua University, Beijing, China
[4] Zhejiang Provincial Key Laboratory of Forestry Intelligent Monitoring and Information Technology, Zhejiang A&F University, Hangzhou, China
[5] College of Mathematics and Computer Science, Zhejiang A&F University, Hangzhou, China

Corresponding author
Liuchang Xu, xuliuchang@zafu.edu.cn

## ABSTRACT

Melody and lyrics, reflecting two unique human cognitive abilities, are usually combined in music to convey emotions. Although psychologists and computer scientists have made considerable progress in revealing the association between musical structure and the perceived emotions of music, the features of lyrics are relatively less discussed. Using linguistic inquiry and word count (LIWC) technology to extract lyric features in 2,372 Chinese songs, this study investigated the effects of LIWC-based lyric features on the perceived arousal and valence of music. First, correlation analysis shows that, for example, the perceived arousal of music was positively correlated with the total number of lyric words and the mean number of words per sentence and was negatively correlated with the proportion of words related to the past and insight. The perceived valence of music was negatively correlated with the proportion of negative emotion words. Second, we used audio and lyric features as inputs to construct music emotion recognition (MER) models. The performance of random forest regressions reveals that, for the recognition models of perceived valence, adding lyric features can significantly improve the prediction effect of the model using audio features only; for the recognition models of perceived arousal, lyric features are almost useless. Finally, by calculating the feature importance to interpret the MER models, we observed that the audio features played a decisive role in the recognition models of both perceived arousal and perceived valence. Unlike the uselessness of the lyric features in the arousal recognition model, several lyric features, such as the usage frequency of words related to sadness, positive emotions, and tentativeness, played important roles in the valence recognition model.

## INTRODUCTION

The pursuit of emotional experience is a vital motivation for listening to music (*Juslin & Sloboda, 2001*; *Juslin & Laukka, 2004*), and the ability to convey emotions ensures the

important role of music in human life (*Yang, Dong & Li, 2018*). Therefore, the relationship between music and perceived emotional expression has attracted increasing academic attention in recent decades (*Swaminathan & Schellenberg, 2015*). Most of the related studies have focused on investigating the association between musical structure and perceived emotions. For example, psychologists have made considerable progress in revealing structural factors (*e.g.*, tempo, pitch, and timbre), indicating different emotional expressions (*Gabrielsson, 2016*), and computer scientists have focused on extracting features from audio (audio most commonly refers to sound, as it is transmitted in signal form; *e.g.*, Mel-frequency cepstrum coefficients and Daubechies wavelet coefficient histograms) to automatically identify music emotion (*Yang, Dong & Li, 2018*). Previous works have shown that sound features were highly correlated with music emotions (*Gabrielsson, 2016*; *Yang, Dong & Li, 2018*), but the lyric features have been relatively less discussed. *Besson et al. (1998)* proved that melodic and lyrical components in music are processed independently. Although melodic information may be more dominant than lyrics in conveying emotions (*Ali & Peynircioğlu, 2006*), investigating the relationship between lyrical structure and the perceived emotion of music in detail is still necessary.

Music emotion studies related to lyrics have often focused on investigating the differences between the presence and absence of lyrics (*Ali & Peynircioğlu, 2006*; *Brattico et al., 2011*; *Yu et al., 2019a*; *Yu et al., 2019b*), the effects of consistency or differences in melodic and lyrical information (*Morton & Trehub, 2001*; *Vidas et al., 2020*), or the effects of lyrics with different meanings (*Batcho et al., 2008*; *Stratton & Zalanowski, 1994*). While lyric structures and features have been rarely studied, melodic information has been processed in previous psychology studies (*Gabrielsson, 2016*; *Swaminathan & Schellenberg, 2015*; *Xu et al., 2020a*). On the other hand, with the development of natural language processing (NLP) technology, different lyric features have been widely extracted and analyzed in music emotion recognition (MER) studies (*e.g.*, *Malheiro et al., 2016a*; *Malheiro et al., 2016b*; *Delbouys et al., 2018*), a field investigating computational models for detecting music emotion (*Aljanaki, Yang & Soleymani, 2017*; *Chen et al., 2015a*). These MER studies have typically focused on improving the prediction effect of constructed models but have not interpreted the model and variables. Thus, can the structural factors of lyrics be analyzed in more detail by combining NLP technology? If so, this may facilitate the understanding of the relationship between lyrics and perceived emotions. Therefore, the present study investigated the effects of various lyric features on the perceived emotions in music.

## RELATED WORK

### Emotion perception in music with lyrics

Melody and lyrics, reflecting two unique human cognitive abilities, are usually combined in music to convey various information (*Gordon et al., 2010*). A melody is a linear succession of musical tones that the listener perceives as a single entity (*van Waesberghe, 1955*); and lyrics is the composition in verse which is sung to a melody to constitute a song. *Besson et al. (1998)* and *Bonnel et al. (2001)* have shown that the melodic and lyrical components are processed independently. However, they are often integrated in such a way that melodic

information is enhanced by the simultaneous presence of lyrics (*Serafine, Crowder & Repp, 1984*; *Serafine et al., 1986*). Studies have been continuously updated to investigate the interaction or independence of melody and lyrics by using melody only (*Bonnel et al., 2001*; *Brattico et al., 2011*; *Kolinsky et al., 2009*), lyrics only (*Fedorenko et al., 2009*; *Poulin-Charronnat et al., 2005*; *Yu et al., 2019b*), or both (*Gordon et al., 2010*; *van Besouw, Howard & Ternström, 2005*). The role of melody and lyrics in music emotion perception is a vital research focus.

As the soul of music, perceived emotion has been widely discussed in recent decades (*Swaminathan & Schellenberg, 2015*). *Ali & Peynircioğlu (2006)* investigated differences in melodies and lyrics conveying the same and mismatched emotions and confirmed the dominance of melody in music emotional information processing. Additionally, they observed that lyrics can strengthen the perception of negative emotions but weaken the perceived positive emotions. A computational study (*Hu, Downie & Ehmann, 2009*) found that negative emotion classification accuracy was improved by adding lyric information, while the opposite effect was obtained for the classification of positive emotions. In contrast, the results of *Laurier, Grivolla & Herrera (2008)* showed that lyrics can facilitate the recognition of happy and sad musical emotions but not angry and violent emotions. Explanatory studies were also conducted instantly following the observed phenomena. *Pieschl & Fegers (2016)* advocated using short-term effects on cognition and affect to explain the power of lyrics. By comparing music with and without lyrics, evidence from functional magnetic resonance imaging also indicated the importance of lyrics for negative musical emotions (*Brattico et al., 2011*). Following the work of *Brattico et al. (2011)*, neural mechanisms have been continually studied in recent years (*e.g.*, *Greer et al., 2019*; *Proverbio, Benedetto & Guazzone, 2020*). In sum, although subtle conflicts exist in different studies, the substantial role of lyrics in music emotion perception is consistent.

The aforementioned theoretical findings have also been supplemented or utilized in other fields. For instance, developmental psychology studies have proven that lyrical information dominates children's judgment of music emotion (*Morton & Trehub, 2001*; *Morton & Trehub, 2007*), but adults rely on melody (*Ali & Peynircioğlu, 2006*; *Vidas et al., 2020*); music therapy studies have widely conducted lyric analyses to extend the understanding of clients' emotional and health states (*Baker et al., 2009*; *O'Callaghan & Grocke, 2009*; *Silverman, 2020*; *Viega & Baker, 2017*); and in computational studies, lyrical information used as additional inputs can significantly improve the predictive effects of MER models (*Laurier, Grivolla & Herrera, 2008*; *Malheiro et al., 2018*; *Yu et al., 2019a*). These studies provide a development direction and practical value of basic lyrics research and encourage the optimization of basic research.

One of the limitations in previous behavioral research is that the lyric was always treated as a complete object. Studies usually investigated the differences between the presence and absence of lyrics (*Ali & Peynircioğlu, 2006*; *Yu et al., 2019b*) or the effects of lyrics with a certain meaning (*e.g.*, lyrics expressing homesickness in *Batcho et al., 2008*, or happy and sad lyrics in *Brattico et al., 2011*) but rarely analyzed the elements extracted from lyrics. In melody-related research, various musical structural factors (*e.g.*, mode, timbre, tempo, and pitch) have been studied (*Juslin & Laukka, 2004*), and the association between

perceived emotion and structural factors has been repeatedly verified in recent decades (*Gabrielsson, 2016*; *Hunter & Schellenberg, 2010*). Therefore, to better understand the relationship between lyrics and music perception emotions, can similar methods be used to analyze the structural factors of lyrics? In addition, unlike musical structural factors, which have been summarized in musicology, analyzing lyrical factor types remains a challenge. We noticed that recent linguistic and NLP-based computational studies can provide inspiration.

## NLP-based lyric features

NLP technology has been widely used to analyze the emotions expressed or perceived in texts, such as book reviews (*Zhang, Tong & Bu, 2019*), opinions on social media platforms (*Xu et al., 2021a*), movie reviews (*Kaur & Verma, 2017*; *Lu & Wu, 2019*), party statements (*Haselmayer & Jenny, 2017*), and song lyrics (*Rachman, Samo & Fatichah, 2019*). Knowledge-based approaches and machine learning-based approaches are two common approaches used for emotional analysis (*Liu & Chen, 2015*). The knowledge-based approach is an unsupervised approach that uses an emotional dictionary or lexicon (a list of words or expressions used to express human emotions) to label emotional words in text (*Liu, 2012*). Thus, a high-quality emotional dictionary is the basis of this approach. In contrast, the machine learning approach is usually a supervised approach that requires a labeled dataset to construct emotion recognition models (*Peng, Cambria & Hussain, 2017*). It usually constitutes a process of (a) extracting text features (including lexical features, syntactic features and semantic features), (b) using machine learning algorithms to construct the relationship between extracted features and labeled emotions, and (c) predicting the emotions of untagged texts.

When conducting emotion analyses of song lyrics, machine learning-based approaches have been more prevalent in the past two decades. *Laurier, Grivolla & Herrera (2008)* used lyric feature vectors based on latent semantic analysis (LSA) dimensional reduction and audio features to conduct music mood classification. They found that standard distance-based methods and LSA were effective for lyric feature extraction, although the performance of lyric features was inferior to that of audio features. *Petrie, Pennebaker & Sivertsen (2008)* conducted linguistic inquiry and word count (LIWC) analyses to explore the emotional changes in Beatles' lyrics over time. *Panda et al. (2013)* used support vector machines, K-nearest neighbors, and naïve Bayes algorithms to map the relationship between music emotion and extracted audio and lyric features. In a recent study, *Zhou, Chen & Yang (2019)* applied unsupervised deep neural networks to perform feature learning and found that this method performed well for audio and lyric data and could model the relationships between features and music emotions effectively. Notably, traditional MER research focuses on improving the prediction effect of the MER models, while our study attempted to use an interpretable way to investigate the relationship between lyrics features and music emotions.

Previous studies have shown a variety of methods for extracting lyric features. Although the lyric feature vectors and the deep learning-based features performed well in MER studies, the meaning of these features is often difficult to understand. Thus, considering

the interpretability of lyric features, this study selected the LIWC-based method to extract lyric features. The LIWC software package was first developed to analyze text for more than 70 language dimensions by *Pennebaker, Francis & Booth (2001)*. It has been applied for text analysis in psychological health (*Slatcher & Pennebaker, 2006*), physical health (*Pennebaker, 2004*), and lyric studies (*Petrie, Pennebaker & Sivertsen, 2008*; *Pettijohn & Sacco, 2009*). The simplified Chinese version of LIWC (SC-LIWC; *Gao et al., 2013*; *Zhao et al., 2016*), which expanded text features for more than 100 dimensions, was also developed in recent years. This technology was considered for lyric feature extraction in this study.

## The present research

In sum, the present study investigates the association between LIWC-based lyric features and the perceived emotions of Chinese songs. First, the direct relationships between the independent variables (lyric features) and the dependent variables (perceived emotions of music) are investigated through correlation analysis. Then, a computational modeling method is considered to examine the effects of lyric features on music emotion perception. Since melody and lyrics are inseparable in music, we use the audio and lyric features extracted in music to predict the perceived emotions. By comparing the prediction effects of the models that use lyric features as input and that lack lyric features, we can intuitively witness the effect of lyrics. Moreover, using interpretable and nonlinear machine learning methods to construct prediction models, different forms of association between lyrics and perceived emotions can be observed (*Vempala & Russo, 2018*; *Xu et al., 2020a*). Finally, the constructed MER models are also of practical value because the recognized music emotion information can be used in various fields, such as music recommendation (*Deng et al., 2015*), music information retrieval (*Downie, 2008*), and music therapy (*Dingle et al., 2015*).

## MATERIALS & METHODS

### Data collection

The original music files and emotion annotation results were obtained from the PSIC3839 dataset, a public-free dataset for MER studies (*Xu et al., 2020b*, unpublished data). In this dataset, arousal and valence scores of 3839 songs popular in China were manually annotated by 87 university students using 5-point Likert scales. Based on the multi-dimensional emotion space model, *Lang (1995)* suggested that emotions can be categorized in a two-dimensional space by valence and arousal; valence ranges from negative to positive, and arousal ranges from passive (low) to active (high). In the PSIC3839 dataset, valence was evaluated from -2 (*negative*) to 2 (*positive*), and arousal was evaluated from -2 (*not at all*) to 2 (*very much*). We then downloaded the lyrics of songs from NetEase Cloud Music (https://music.163.com/), a popular music site in China. Considering that the annotators of the PSIC3839 dataset are all native Chinese, only 2,372 songs with Chinese lyrics were retained for subsequent analysis.

### Lyric feature extraction

To extract the lyric features, the raw data need to be preprocessed. First, since the raw data of lyrics were downloaded online, they contained a large amount of unwanted information,

such as singer, composer, and lyricist names and the words "man" and "woman" in duet songs. Thus, we manually filtered these unwanted information elements. Second, unlike English texts that are directly composed of separated words, Chinese texts require special tools to divide them into separate words for analysis. For example, the sentence "he feels happy" in English text is "hefeelshappy" in Chinese text. Therefore, this study used the Chinese word segmentation tool in the Language Technology Platform (*Che, Li & Liu, 2010*) for text segmentation. Through the above steps, the raw lyrics of each song were processed into Chinese words arranged in order.

After the above data preprocessing, we used SC-LIWC (*Gao et al., 2013*; *Zhao et al., 2016*) to extract the lyric features. A total of 98 types of lyric features were calculated for each song, such as the total number of words (*WordCount*), the proportion of positive emotion words (*PosEmo*), and the proportion of swear words (*Swear*). For example, the lyric feature *PosEmo*, reflecting the usage frequency of positive emotion words in each song, is calculated by dividing the number of positive emotion words in SC-LIWC by the total number of words. All the extracted features are listed and introduced in Supplemental Materials Table S1.

## Audio feature extraction

For audio features, this study considered both rhythmic features (by beat and tempo detection) and spectral features related to timbre, pitch, harmony, and so forth (*e.g.*, Mel-frequency cepstrum coefficients, MFCCs). Audio signal preprocessing was first conducted by (a) using a 22,050 Hz sampling rate to sample each song and (b) using a short-term Fourier transform to obtain the power spectrogram. Then, using the librosa toolkit (*McFee et al., 2015*), a total of nine low or middle features were extracted, including MFCCs, spectral centroid, spectral bandwidth, spectral roll-off, spectral flatness, spectral contrast, tonal centroid features (tonnetz), chromagram, and tempo. Different spectral features were calculated in different ways. For instance, MFCCs were calculated by selecting the lower cepstral coefficients of a Mel-scaled spectrogram, which was generated by using a Mel filter bank to filter the spectrogram (*Meyer & Kollmeier, 2009*).

Since the extracted features of each song were represented in a subspace of high dimensionality, we conducted feature reduction of each type of feature to reduce the storage and computational space. Principal component analysis (PCA), widely used in MER studies (*Xu et al., 2020a*; *Yang, Dong & Li, 2018*), was applied to reduce the dimensionality of features. After conducting the PCA, we selected and combined the top 50 dimensions of each type of audio feature as model inputs. Finally, we scaled each continuous audio feature to a value of 0 to 1 *via* min-max scaling (*Kahng, Mantik & Markov, 2002*). The above processed features were used as the final audio inputs of the computational models.

## Construction of computational models

The proposed modeling method is shown in Fig. 1. Since the annotation results of the perceived valence and arousal values are continuous variables, this study formulated the construction of computational models as a regression problem, which predicts a real value from observed features (*Sen & Srivastava, 2012*). We used the audio and lyric features as

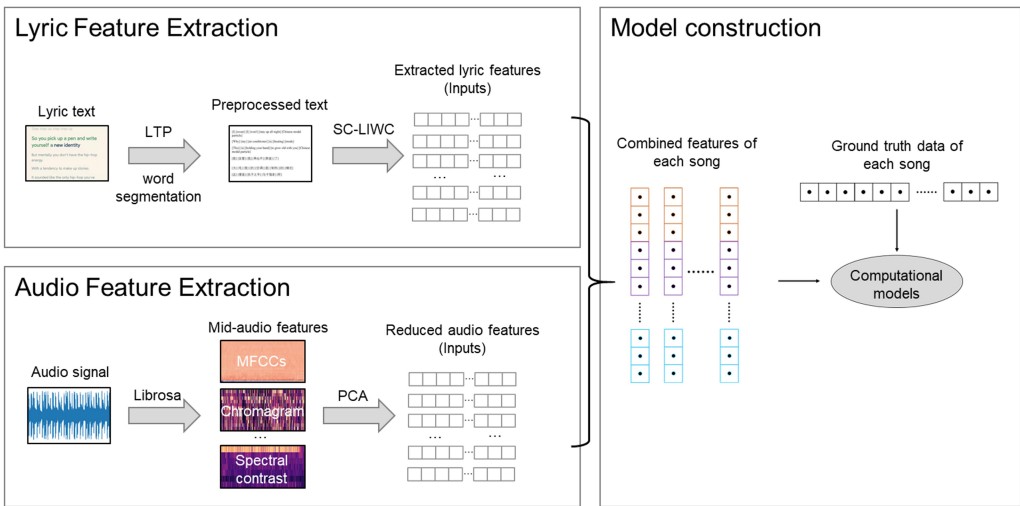

**Figure 1  The proposed modeling method.**

model inputs and the perceived emotion values as ground truth. To explore the effects of lyric features, three types of input sets were considered: (a) audio features only; (b) lyric features only; and (c) combining audio and lyric features. In addition, the ground truth values were also scaled to a value of 0 to 1 *via* min-max scaling before modeling.

Two machine learning algorithms were then considered to map the inputs and perceived emotion values (ground truth data). Multiple linear regression (MLR) was used as the baseline algorithm. Random forest regression (RFR), showing good performance in MER tasks (*e.g.*, *Xu et al., 2020a*; *Xu et al., 2021b*), was used as the main algorithm. For each RFR model, we used a grid parameter search to obtain the best modeling parameters. The performances of our models were evaluated by the tenfold cross-validation technique. The prediction accuracy of each regressor was measured by $R^2$ statistics and the root mean-squared error (RMSE) as follows:

$$R^2 = 1 - \frac{\sum_{i=1}^{N} (X_i - Y_i)^2}{\sum_{i=1}^{N} (X_i - \bar{X})^2}$$

$$RMSE = \sqrt{\frac{1}{N} \sum_{i=1}^{N} (X_i - Y_i)^2}$$

where $X_i$ is the perceived emotion value (ground truth) of each song, $\bar{X}$ is the mean value of the perceived emotion values, $Y_i$ is the predicted result of each song, and $N$ is the number of testing samples in the tenfold cross-validation technique.

# RESULTS

## Data distribution

As the first step in exploring the data, we created a preliminary description of the emotional annotation results of all songs and the content of the lyrics. Figure 2A shows the distribution

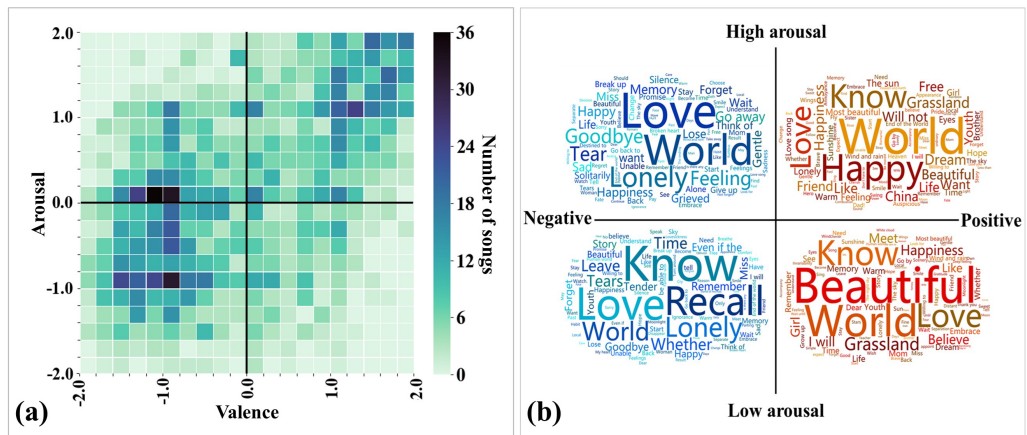

**Figure 2** **Data distribution of the 2372 Chinese songs in this study.** (A) Distribution of annotated emotions in the valence-arousal emotion space. (B) Word clouds of the top words used in each quadrant of the valence-arousal emotion space. The font size depends on the usage frequency of the word (positive correlation).

of annotated emotions of songs in the valence-arousal emotion space. We observed that a large proportion of the songs fell under the third quadrant (37.94%; low arousal and negative), followed by the first (34.49%; high arousal and positive) and fourth quadrants (17.07%; high arousal and negative). Pearson correlation analysis shows that the perceived arousal values are positively correlated with valence (r(2371) = 0.537, $p < .001$).

We then calculated the usage frequency of different words in song sets with different emotions. Excluding commonly used personal pronouns (*e.g.*, "I" and "you") and verbs (*e.g.*, "is" and "are"), the top used words are presented in the word clouds (see Fig. 2B). We observed that the words "love" and "world" frequently appear in every quadrant of the valence-arousal emotion space, meaning that these are two popular song themes. The words "happy" and "beautiful" frequently appear in positive songs, whereas the words "lonely" and "recall" frequently appear in negative songs. The above results allow us to intuitively see the difference in word usage in different emotional songs.

## Correlation analysis between perceived emotions and lyric features

In this part, we analyzed how well the independent variables (lyric features) accounted for the dependent variables (perceived arousal and valence values). The lyric features most relevant to arousal and valence are shown in Tables 1 and 2, respectively. For instance, using Pearson correlation analysis, we found that the perceived arousal values were positively correlated with the total number of words in songs (*WordCount*, r(2371) = 0.206, $p < .01$), the mean number of words per sentence (*WordPerSentence*, r(2371) = 0.179, $p < .01$), the ratio of Latin words (*RateLatinWord*, r(2371) = 0.183, $p < .01$), and the proportion of words related to achievement (*Achieve*, r(2371) = 0.111, $p < .01$) and were negatively correlated with the proportion of words related to the past (*tPast*, r(2371) = −0.124, $p < .01$) and the proportion of words related to insight (*Insight*, r(2371) = −0.122, $p < .01$). For valence, we observed that the perceived valence values were negatively correlated with the proportion

**Table 1  Correlation between lyric features and perceived arousal in music.**

| | 1 | 2 | 3 | 4 | 5 | 6 | 7 | 8 | 9 |
|---|---|---|---|---|---|---|---|---|---|
| 1. Arousal | 1 | | | | | | | | |
| 2. *PastM* | -.121[**] | 1 | | | | | | | |
| 3. *Insight* | -.122[**] | .168[**] | 1 | | | | | | |
| 4. *Time* | -.115[**] | .299[**] | .139[**] | 1 | | | | | |
| 5. *Achieve* | .111[**] | 0.023 | .146[**] | −0.006 | 1 | | | | |
| 6. *tPast* | -.124[**] | .517[**] | .153[**] | .322[**] | 0.014 | 1 | | | |
| 7. *WordCount* | .206[**] | .056[**] | .080[**] | 0.004 | .073[**] | 0.014 | 1 | | |
| 8. *WordPerSentence* | .179[**] | .062[**] | .105[**] | 0.021 | .088[**] | 0.025 | .873[**] | **1** | |
| 9. *RateLatinWord* | .183[**] | −0.029 | 0.02 | −0.032 | .103[**] | −0.031 | .174[**] | .114[**] | 1 |

Notes.
[**]Correlation is significant at the 0.01 level (2-tailed).
Abbreviations: *PastM*, proportion of past tense markers; *Insight*, proportion of words related to insight; *Time*, proportion of words related to time; *Achieve*, proportion of words related to achievement; *tPast*, proportion of words related to the past; *WordCount*, the total number of words; *WordPerSentence*, average number of words per sentence; *RateLatinWord*, the ratio of Latin words.

**Table 2  Correlation between lyric features and perceived valence in music.**

| | 1 | 2 | 3 | 4 | 5 | 6 | 7 | 8 | 9 |
|---|---|---|---|---|---|---|---|---|---|
| 1. Arousal | 1 | | | | | | | | |
| 2. *PastM* | -.121[**] | 1 | | | | | | | |
| 3. *Insight* | -.122[**] | .168[**] | 1 | | | | | | |
| 4. *Time* | -.115[**] | .299[**] | .139[**] | 1 | | | | | |
| 5. *Achieve* | .111[**] | 0.023 | .146[**] | −0.006 | 1 | | | | |
| 6. *tPast* | -.124[**] | .517[**] | .153[**] | .322[**] | 0.014 | 1 | | | |
| 7. *WordCount* | .206[**] | .056[**] | .080[**] | 0.004 | .073[**] | 0.014 | 1 | | |
| 8. *WordPerSentence* | .179[**] | .062[**] | .105[**] | 0.021 | .088[**] | 0.025 | .873[**] | 1 | |
| 9. *RateLatinWord* | .183[**] | −0.029 | 0.02 | −0.032 | .103[**] | −0.031 | .174[**] | .114[**] | 1 |

Notes.
[**]Correlation is significant at the 0.01 level (2-tailed).
Abbreviations: *Adverb*, proportion of adverbs; *TenseM*, proportion of tense markers; *PastM*, proportion of past tense markers; *NegEmo*, proportion of negative emotion words; *Sad*, proportion of words related to sadness; *CogMech*, proportion of words related to cognition; *Tentat*, proportion of words related to tentativeness; *tPast*, proportion of words related to the past.

of negative emotion words (*NegEmo*, $r(2371) = −0.364$, $p < .01$) and proportion of words related to sadness (*Sad*, $r(2371) = −0.299$, $p < .01$). The entire correlation results are presented in Supplemental Materials TabelS2. The correlation results only reveal the linear relationships between perceived emotions and lyric features. Thus, we then used machine learning methods to investigate other types of relationships.

## Model prediction results

In this section, we used MLR and RFR algorithms, different input sets (audio features, lyric features, and combined features), and different ground truth data (arousal and valence) to construct MER models. To obtain relatively good models, a grid parameter search was first conducted to obtain the best performing parameters for each RFR model (results are shown in Table 3).

**Table 3** The best performing parameters for each random forest regression.

| Ground truth | Inputs | Parameters | | | | |
|---|---|---|---|---|---|---|
| | | *n_estimators* | *max_depth* | *min_samples_leaf* | *min_samples_split* | *max_features* |
| | AF | 156 | 10 | 8 | 18 | 0.2 |
| Arousal | LF | 196 | 50 | 5 | 8 | 0.8 |
| | CF | 136 | 27 | 3 | 12 | 0.4 |
| | AF | 179 | 15 | 13 | 22 | 0.5 |
| Valence | LF | 193 | 38 | 3 | 8 | 0.8 |
| | CF | 191 | 43 | 5 | 25 | 0.6 |

Notes.
Abbreviations: AF indicates audio features; LF indicates lyric features; and CF indicates combined features.

After parameter searching, the performances of the constructed models, evaluated by the tenfold cross-validation technique, are presented in Fig. 3. Since the tenfold cross-validation technique, using 10% data as the testing data and using the remaining 90% instances as training data to train regressor, used the same folds in the evaluation of different models, a tenfold paired sample $t$-test can be applied to compare the model results. For the algorithms, the RFR algorithm performed better than the MLR algorithm in all the constructed models. For example, ten-fold paired sample $t$-test showed that, using combined features to predict perceived arousal values, the RFR-based model reached a mean $R^2$ value of 0.631 and a mean RMSE value of 0.147, significantly better than the MLR-based model ($R^2 = 0.532$, $t(9) = 4.206$, $p < .01$, $d = 1.883$; RMSE $= 0.165$, $t(9) = -4.012$, $p < .01$, $d = -1.960$). Therefore, the subsequent analysis was only conducted on the models based on the RFR algorithm.

For the recognition models of perceived arousal values, paired sample $t$-test showed that the model using audio features as inputs performed significantly better than the model using lyric features ($R^2$: $t(9) = 36.335$, $p < .001$, $d = 15.219$; RMSE: $t(9) = -34.693$, $p < .001$, $d = -12.167$). Although the model using combined features ($R^2 = 0.631$, RMSE $= 0.147$) performed slightly better than the model using audio features ($R^2 = 0.629$, RMSE $= 0.147$), there was no significant effect ($R^2$: $t(9) = 0.134$, $p = .896$, $d = 0.063$; RMSE: $t(9) = -0.231$, $p = .822$, $d = -0.114$). These results revealed that the perceived arousal of music mainly depends on audio information, while lyrics are not important.

For valence, paired sample $t$-test showed that, although the best performing recognition model of perceived valence values performed worse than the arousal model ($R^2$: $t(9) = -6.700$, $p < .001$, $d = -3.271$; RMSE: $t(9) = 11.539$, $p < .001$, $d = 6.094$), both lyric and audio features played important roles. The RFR-based model using audio features only reached a mean $R^2$ value of 0.371 and a mean RMSE value of 0.214, and the model using lyric features as inputs reached a mean $R^2$ value of 0.370 and a mean RMSE value of 0.214. When the audio and lyric features were combined as inputs, the new model achieved a mean $R^2$ value of 0.481 and a mean RMSE value of 0.194, significantly better than the previous two models. These results indicated that the perceived valence of music was influenced by both audio and lyric information.

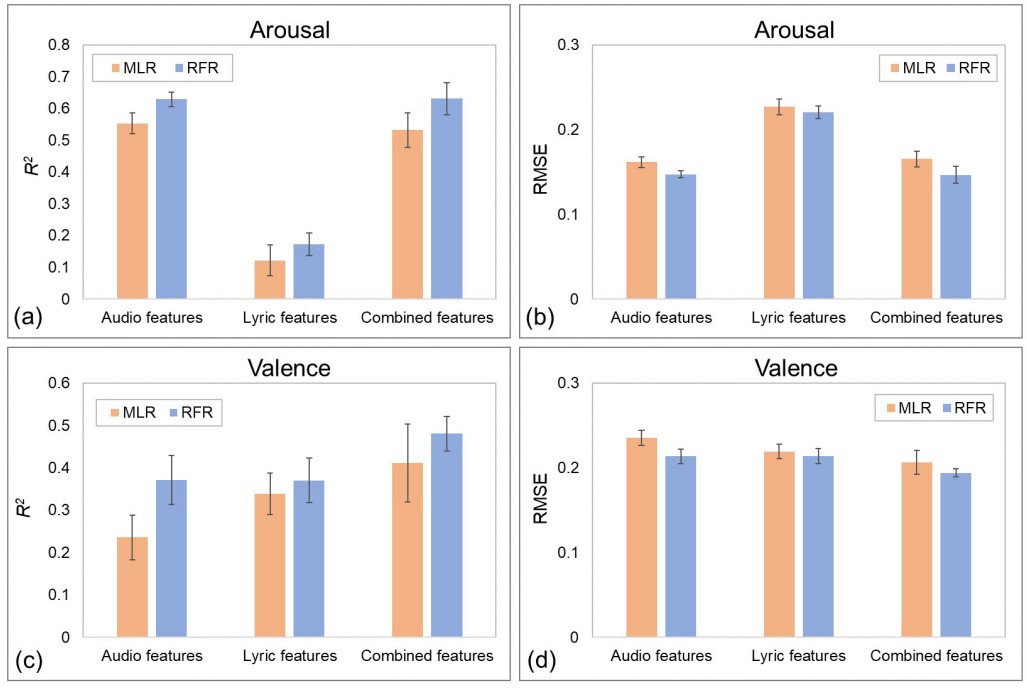

**Figure 3** **Performance of constructed MER models with different inputs and algorithms.** (A) Prediction results of perceived arousal recognition models, measured by $R^2$ statistics. (B) Prediction results of perceived arousal recognition models, measured by *RMSE*. (C) Prediction results of perceived valence recognition models, measured by $R^2$ statistics. (D) Prediction results of perceived valence recognition models, measured by *RMSE*. Error bars indicate the standard deviations.

## Model interpretability

In the last step, we attempted to explain the best performing RFR-based models by examining the information gain of features. Quiroz, Geangu, & Yong (2018) noted that models constructed using the RFR algorithm can be interpreted by calculating feature importance. Thus, the feature importance of the best performing recognition models of the perceived arousal and valence was calculated and is presented in Fig. 4. Since tenfold cross-validation was used to evaluate the models, the coefficients of feature importance might differ when predicting different test sets (*Xu et al., 2020a*). Thus, the distribution of feature importance was arranged in descending order of the mean value, and only the top 30 features were included for visibility.

In the arousal recognition model, the audio features played a decisive role, which accounted for 95.01% of the model. The first PCA components of spectral flatness, spectral contrast, chromagram, MFCCs, and spectral bandwidth are the five most contributing features, accounting for 30.71%, 12.38%, 8.77%, 8.62%, and 4.26%, respectively. For the lyric features, the feature importance results are similar to the results of the correlation analysis. The total number of words in songs (*WordCount*) and the mean number of words per sentence (*WordPerSentence*) were the top two contributing lyric features, accounting for 0.19% and 0.15% of the model.

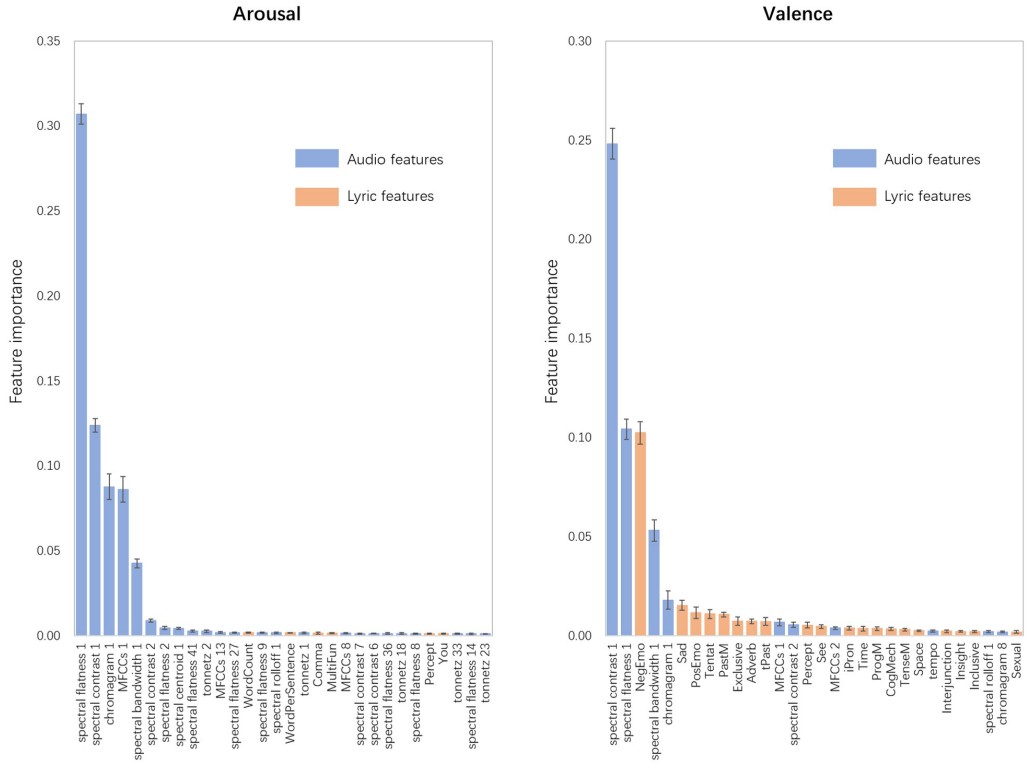

**Figure 4** **Distribution of feature importance for RFR-based recognition models of perceived arousal and perceived valence.** Arranged in descending order of the mean value, the top 30 features were included for visibility, and the trend of the remaining features was approximately the same. Error bars indicate the standard deviations.

For valence, the audio features explained 73.44% of the model. The first PCA components of spectral contrast and spectral flatness also showed good predictive effects on the perceived valence, accounting for 24.82% and 10.42% of the model, respectively. The proportion of negative emotion words (*NegEmo*) was the most important lyric feature (accounting for 5.32%), followed by the proportion of words related to sadness (*Sad*, 0.56%), positive emotions (*PosEmo*, 0.39%), tentativeness (*Tentat*, 0.23%), and so on. These findings also support the opinion that lyric features can provide more information for recognition models of valence than for recognition models of arousal.

## DISCUSSION

This study investigated the effects of LIWC-based lyric features on the perceived arousal and valence of music. We first explored the data distribution, and several interesting results were found. First, the emotional distribution (in the valence-arousal emotion space) of music in this study is similar to previous works. Various studies have shown that the perceived valence and arousal of music were positively correlated (*e.g.*, *Chen et al., 2015b*; *Greenberg et al., 2016*; *Speck et al., 2011*). This reveals that the relationship between valence and arousal in music is relatively constant. Second, analyzing the songs' word usage

frequency in different quadrants of the valence-arousal emotion space, we then found that the word "love" frequently appears in each quadrant. The cross-cultural study of *Freeman (2012)* has shown that "romantic love" is the top topic category of Chinese-language pop songs (82.5%), while Western pop songs with the topic of "romantic love" accounted for only 40%. The high usage frequency of the word "love" in this study further confirms that love-themed songs are the mainstream in Chinese pop music. In addition, we observed that the words "happy" and "beautiful" frequently appear in positive songs, whereas the words "lonely" and "recall" frequently appear in negative songs. This intuitive phenomenon shows that, in general, perceived music emotions are related to lyrics, which encourages us to further explore lyric features.

Although LIWC-based lyric features have been considered in computational studies to improve the model effect (*e.g.*, *Hu, Chen & Yang, 2009*; *Malheiro et al., 2016a*; *Malheiro et al., 2016b*), the role of LIWC-based lyric features has never been analyzed and discussed individually. Thus, this study then investigated the linear relationship between each lyric feature and the perceived emotion of music. In general, valence is more correlated with the features reflecting the meaning of the lyric text, while arousal is more correlated with the features reflecting the structure of the lyrics. For example, we observed that the perceived valence values were negatively correlated with the usage frequency of words related to negative emotions (*e.g.*, "sad", "horrible", "angry", and "bitter"), tentativeness (*e.g.*, "seem", "dim", "guess", and "dubitation"), insight (*e.g.*, "understanding", "notice", "analyze", and "know"), and exclusiveness (*e.g.*, "exclude", "forget", "ignore", and "cancel"). It is obvious that the usage of emotional words is related to perceived emotions because emotional words shape emotional percepts (*Gendron et al., 2012*). Words related to tentativeness are often used in sad love songs to express doubts about love (*e.g.*, "In the next few days, I guess you won't show up either"), and insight words are often used with negative words to portray the sad atmosphere (*e.g.*, "no one notices me, only a small raindrop accompanies me to wait for dawn"). We believe that some words in lyrics are often used to describe certain behaviors, feelings or scenes, which are related to negative emotions. For instance, nostalgia, characterized by sadness, insomnia, loss of appetite, pessimism, and anxiety (*Batcho et al., 2008*; *Havlena & abd Holak, 1991*), is one of the themes of songs. Thus, words that describe nostalgia in lyrics are often related to negative emotions. This phenomenon may appear in songs with various themes, such as farewell, war, and tragic love stories.

For arousal, we observed that the perceived arousal values were positively correlated with the total number of lyric words (*WordCount*) and the mean number of words per sentence (*WordPerSentence*). We wonder that the size of music may play an important role, because the duration of music is positively correlated with the total number of lyric words ($r(2371) = 0.219$, $p < .01$); that is, the longer the music, the more words are in the lyrics. Unfortunately, as shown in Supplemental Materials Fig. S1, the duration of music was negatively correlated with the arousal values ($r(2371) = −0.111$, $p < .01$). In addition, *Holbrook & Anand (1990)* found that the tempo of music is positively correlated with listeners' perceived arousal. Thus, another assumption is that fast-paced songs tend to match more lyric words. Unfortunately, the above hypothesis was not confirmed in

the current data set (r(2371) = −0.039, $p$ = .058). This result reminds us to analyze the relationship between audio features and the number of words. We found that the total number of lyric words was positively correlated with chromagram and negatively correlated with MFCCs (see Supplemental Materials Table S3). Chromagrams reflect the pitch components of music over a short time interval (*Schmidt, Turnbull & Kim, 2010*). Previous work on screams has found a significant tendency to perceive higher-pitched screams as more emotionally arousing than lower-pitched screams (*Schwartz & Gouzoules, 2019*). However, whether the above phenomenon holds in music is still unknown, and it is worthy of further research. While MFCCs reflect the nonlinear frequency sensitivity of the human auditory system (*Wang et al., 2012*), it is difficult to map well-known music features in conventional musical writing. The low-level audio features in this study may not directly explain the relationship between melody features and lyrics. In fact, *McVicar, Freeman & De Bie (2011)* found it hard to interpret the correlations between arousal and lyric features. Therefore, how to map the relationships among arousal, melody, and lyrics still needs further investigation.

The above correlation analysis reflects the direct connection between lyric features and perceived emotions. We then used audio features and lyric features to construct MER models. By comparing the results of models and calculating feature importance to interpret the constructed models, we investigated the role of lyric features and obtained two major discoveries. First, we found that, compared with lyric features, audio features played a decisive role in the MER models for both perceived arousal and perceived valence. From the perspective of computational modeling, this finding confirms previous conclusions that melodic information may be more dominant than lyrics in conveying emotions (*Ali & Peynircioğlu, 2006*). However, previous works used individual moods affected by music to evaluate the ability of music to convey emotions (*Ali & Peynircioğlu, 2006*; *Sousou, 1997*), which is not equivalent to the perceived emotions of music. Thus, our study provided more direct evidence that melody information plays a decisive role in the perception of music emotions, and we believe that this result can be generalized to all countries. The second major finding was that, unlike the uselessness of the lyric features in the arousal recognition model, lyric features can significantly improve the prediction effect of the valence recognition model. Feature importance analysis also shows that lyric features, such as the proportion of words related to sadness (*Sad*), positive emotions (*PosEmo*), and tentativeness (*Tentat*), played important roles in the valence recognition model. This finding was consistent with that of *Hu, Downie & Ehmann (2009)*, which showed that lyrics can express the valence dimension of emotion but usually do not express much about the arousal dimension of emotion, rather than the opposite finding shown by *Malheiro et al. (2016a)* and *Malheiro et al. (2016b)*. We hypothesize that the main reason for the difference in results is that our study and the study of *Hu, Downie & Ehmann (2009)* both focused on Chinese music and participants, but the study of *Malheiro et al. (2016a)* and *Malheiro et al. (2016b)* was conducted in Portugal. Cross-cultural studies have shown that although listeners are similarly sensitive to musically expressed emotion (which is facilitated by psychophysical cues; (*Argstatter, 2016*; *Balkwill & Thompson, 1999*), differences still exist (*Zacharopoulou & Kyriakidou, 2009*). Therefore, we believe that in the

Chinese environment, perceived music valence is affected by lyrics, although its influence is not as strong as that of melody information.

As mentioned before, this study is also of practical value. The computational modeling method was first proposed in the field of MER, which aims to automatically recognize the perceptual emotion of music (*Yang, Dong & Li, 2018*). There are many existing songs, but it is difficult for people to manually annotate all emotional information. Thus, MER technology is urgently needed and has made great progress in the past two decades. Recognized emotion information can be used in various application scenarios (*Deng et al., 2015*; *Dingle et al., 2015*; *Downie, 2008*). The collected data and proposed methods in this study can also provide references for future MER research. Notably, the computational modeling methods in MER studies pursue model effects and prediction accuracy, but when they are applied in music psychology research, the interpretability of the model should be taken into account (*Vempala & Russo, 2018*; *Xu et al., 2020a*). Therefore, we chose MLR and RFR to construct MER models. How to integrate machine learning methods into music psychology research more effectively still needs more exploration.

## CONCLUSIONS

The present work investigated the effects of LIWC-based lyric features on the perceived arousal and valence of music by analyzing 2372 Chinese songs. Correlation analysis shows that, for example, the perceived arousal of music was positively correlated with the total number of lyric words (*WordCount*, $r(2371) = 0.206$, $p < .01$) and the mean number of words per sentence (*WordPerSentence*, $r(2371) = 0.179$, $p < .01$) and was negatively correlated with the proportion of words related to the past (*tPast*, $r(2371) = -0.124$, $p < .01$) and insight (*Insight*, $r(2371) = -0.122$, $p < .01$). The perceived valence of music was negatively correlated with the proportion of negative emotion words (*NegEmo*, $r(2371) = -0.364$, $p < .01$) and the proportion of words related to sadness (*Sad*, $r(2371) = -0.299$, $p < .01$). We then used audio and lyric features as inputs to construct MER models. The performance of RFR-based models shows that, for the recognition models of perceived valence, adding lyric features can significantly improve the prediction effect of the model using audio features only; for the recognition models of perceived arousal, lyric features are almost useless. Calculating the importance of features to interpret the MER models, we observed that the audio features played a decisive role in the recognition models of both perceived arousal and perceived valence. Unlike the uselessness of the lyric features in the arousal recognition model, several lyric features, such as the proportion of words related to sadness (*Sad*), positive emotions (*PosEmo*), and tentativeness (*Tentat*), played important roles in the valence recognition model.

### Funding

This research was supported by the Research and Development Foundation of Zhejiang A&F University under Grant 2020FR064, the Open Research Fund of Zhejiang Provincial

Key Laboratory of Resources and Environmental Information System under Grant 2020330101004109, and the Scientific Research Foundation of the Education Department of Zhejiang Province, China, under Grant Y202147221. There was no additional external funding received for this study. The funders had no role in study design, data collection and analysis, decision to publish, or preparation of the manuscript.

### Grant Disclosures
The following grant information was disclosed by the authors:
The Research and Development Foundation of Zhejiang A&F University: 2020FR064.
The Open Research Fund of Zhejiang Provincial Key Laboratory of Resources and Environmental Information System: 2020330101004109.
The Scientific Research Foundation of the Education Department of Zhejiang Province, China: Y202147221.

### Competing Interests
The authors declare there are no competing interests.

### Author Contributions
- Liang Xu conceived and designed the experiments, analyzed the data, performed the computation work, prepared figures and/or tables, and approved the final draft.
- Zaoyi Sun conceived and designed the experiments, performed the experiments, authored or reviewed drafts of the paper, and approved the final draft.
- Xin Wen performed the experiments, prepared figures and/or tables, and approved the final draft.
- Zhengxi Huang analyzed the data, prepared figures and/or tables, and approved the final draft.
- Chi-ju Chao performed the computation work, prepared figures and/or tables, and approved the final draft.
- Liuchang Xu analyzed the data, authored or reviewed drafts of the paper, and approved the final draft.

### Data Availability
   The raw data is available in the Supplementary File.

### Supplemental Information
Supplemental information for this article can be found online at http://dx.doi.org/10.7717/peerj-cs.785#supplemental-information.

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
