# Peer review of "Using machine learning analysis to interpret the relationship between music emotion and lyric features"

_PeerJ Computer Science, doi:10.7717/peerj-cs.785_

## Round 0.1 · original submission · Major Revisions

The main issues regarding this manuscript were presented by Reviewer 1. Please address the following major issues and other minor issues
pointed out by reviewers:
- Generalization to different countries
- Explanation of the observed correlations
- Is there a threshold to trigger emotional responses
- Discussion about extensive and intensive features
- Language quality should be improved

Reviewer 1 ·

Basic reporting

In this paper, the authors address the emotional effect of features extracted from song's audio and lyrics on human listeners. In particular, the main goal of the paper is to use NLP techniques to see the role of music's audio/lyrics can raise any emotional response on the listeners.

The paper is well written and clear. The context and background information is supported by literature references. The paper is well structured.

I would like to make the following comments about the paper:

1) The authors could stress the definitions for the "musical" terms used in the paper, such as: melody, lyrics, audio, valence, arousal;

2) How can someone tell if a music is positive or negative without lyric? In the literature, is there a study relating sound features to the emotional response?

3) The authors mention that the results may depend on the country where the study is conducted since the local culture may affect the way an individual interprets the music. In which extent the results presented here may be generalized to all countries?

4) What is the difference between music emotion (Panda et al. 2013) and the idea of emotion perception in music? I think the authors could emphasize how and why this work is different from Panda et al. 2013.

5) Is there an explanation for the correlation between arousal and the total number of words of a song? Do you think that the duration of the music play a significant role? In other words, is there a threshold (minimum number of words / duration) that has to be reached so a music can trigger an emotional response?

6) The authors could explain how the numbers 2371 and 18 (in r(2371) and t(18), respectively) were obtained.

7) The authors could discuss about the use of "extensive" features (that may depend on the song size) instead of "intensive" ones (normalized, that may not depend on the song size attributes such as the number of words). If an emotional response (like arousal) is triggered by the song size, does it mean that a longer song will trigger a more intense emotional response? I wonder this relationship is always linear, in the sense that a song that is too long (too short) will trigger the same kind of response that a medium size song. It would be nice to see a two dimensional histogram showing the intensity of the emotional response and the size of the music.

Experimental design

No comment.

Validity of the findings

No comment.

Reviewer 2 ·

Basic reporting

General

Taking into account the possibility that this manuscript does get published, I think it appropriate to include a few brief comments.
- The manuscript seems somewhat with grammatical/syntax and typographical problems. I leave it to the authors to resolve these copyediting problems by actually thoroughly reading the manuscript. Problems of this sort should definitely not appear in print.

Title
It is concise and conveys the topic of research.

Abstract and Conclusions sections included are basically the same text.

Experimental design

no comment

Validity of the findings

no comment

---

## Round 0.2 · accepted · Accept

All issues have been addressed.

Reviewer 1 ·

Basic reporting

No comment.

Experimental design

No comment.

Validity of the findings

No comment.

Additional comments

The authors addressed the points I raised previously, thus I recommend the publication of the paper.

Reviewer 2 ·

Basic reporting

Thank you for trusting in my work. All proposed suggestions were provided in this last version. I'd like to suggest some proof-editing:
At line 229, including a period at the end of the paragraph
At line 309, include a space between Table and S2 words - Supplemental Materials TabelS2
At line 341, replace the capital letter T for lowercase t
In order to apply a default format of the numbers used in the manuscript, I’d suggest including 0 before all occurrences of the decimals numbers (sometimes is used .01, .111, and 0.01)

Experimental design

no comment

Validity of the findings

no comment

Additional comments

All proposed suggestions were provided in this last version. I'd like to suggest some proof-editing:
At line 229, including a period at the end of the paragraph
At line 309, include a space between Table and S2 words - Supplemental Materials TabelS2
At line 341, replace the capital letter T for lowercase t
In order to apply a default format of the numbers used in the manuscript, I’d suggest including 0 before all occurrences of the decimals numbers (sometimes is used .01, .111, and 0.01)